# Integrated Metabolomics and Morpho-Biochemical Analyses Reveal a Better Performance of *Azospirillum brasilense* over Plant-Derived Biostimulants in Counteracting Salt Stress in Tomato

**DOI:** 10.3390/ijms232214216

**Published:** 2022-11-17

**Authors:** Mónica Yorlady Alzate Zuluaga, Begoña Miras-Moreno, Sonia Monterisi, Youssef Rouphael, Giuseppe Colla, Luigi Lucini, Stefano Cesco, Youry Pii

**Affiliations:** 1Faculty of Science and Technology, Free University of Bozen-Bolzano, 39100 Bolzano, Italy; 2Department for Sustainable Food Process, Research Centre for Nutrigenomics and Proteomics, Università Cattolica del Sacro Cuore, 29122 Piacenza, Italy; 3Department of Agricultural Sciences, University of Naples Federico II, 80055 Portici, Italy; 4Department of Agriculture and Forest Sciences, University of Tuscia, 01100 Viterbo, Italy

**Keywords:** antioxidant defense system, bacterial biostimulant, metabolomic profile, protein hydrolysates, salinity tolerance

## Abstract

Increased soil salinity is one of the main concerns in agriculture and food production, and it negatively affects plant growth and crop productivity. In order to mitigate the adverse effects of salinity stress, plant biostimulants (PBs) have been indicated as a promising approach. Indeed, these products have a beneficial effect on plants by acting on primary and secondary metabolism and by inducing the accumulation of protective molecules against oxidative stress. In this context, the present work is aimed at comparatively investigating the effects of microbial (i.e., *Azospirillum brasilense*) and plant-derived biostimulants in alleviating salt stress in tomato plants by adopting a multidisciplinary approach. To do so, the morphological and biochemical effects were assessed by analyzing the biomass accumulation and root characteristics, the activity of antioxidant enzymes and osmotic stress protection. Furthermore, modifications in the metabolomic profiles of both leaves and root exudates were also investigated by ultra-high performance liquid chromatography/quadrupole time-of-flight mass spectrometry (UHPLC/QTOF-MS). According to the results, biomass accumulation decreased under high salinity. However, the treatment with *A. brasilense* considerably improved root architecture and increased root biomass by 156% and 118% in non-saline and saline conditions, respectively. The antioxidant enzymes and proline production were enhanced in salinity stress at different levels according to the biostimulant applied. Moreover, the metabolomic analyses pointed out a wide set of processes being affected by salinity and biostimulant interactions. Crucial compounds belonging to secondary metabolism (phenylpropanoids, alkaloids and other N-containing metabolites, and membrane lipids) and phytohormones (brassinosteroids, cytokinins and methylsalicylate) showed the most pronounced modulation. Overall, our results suggest a better performance of *A. brasilense* in alleviating high salinity than the vegetal-derived protein hydrolysates herein evaluated.

## 1. Introduction

The excessive concentrations of salts in soils, irrigation water and the massive use of chemical fertilizers are considered the major concerns affecting plant growth and crop productivity [1]. Among the principal salts found in soils (i.e., Na^+^, Cl^−^, Ca^2+^, Mg^2+^, K^+^, SO_2_^4−^, CO^3−^, and NO^3−^), Na^+^ and Cl^−^ have been described as the ions mainly responsible for soil salinization, reaching concentrations that induce several morphological, biochemical, physiological and metabolic responses, hindering the equilibrate plant development [2,3]. Multiple adverse effects are triggered in plants under saline conditions; among these, the most frequently reported include (i) osmotic stress due to the decreased root water uptake, (ii) oxidative stress associated with the enhanced production of the reactive oxygen species (ROS), and (iii) ionic stress ascribable to the toxic effect of salt ions inside plant cells [4].

In the attempt to cope with high salt concentrations, plants have evolved adaptive responses involving different metabolomic, genomic and proteomic pathways [1]. For example, as a primary adaptive strategy, plants synthesize and accumulate compatible osmolytes (e.g., proline, glycine, betaine, polyols, sugars) crucial in osmotic adjustment and detoxification. Furthermore, salt-stressed plants can also induce the biosynthesis of enzymatic ROS detoxification (such as superoxide dismutase—SOD, ascorbate peroxidase—APX, and catalase—CAT), as well as a broad set of flavonoids and phenolic compounds [4]. Moreover, plants can also alter their physiological and morphological features by modulating developmental and growth responses, such as root architecture, blooming time, leaf senescence and biomass allocation [5,6]. However, these adaptive responses alone are not enough to guarantee balanced plant growth and the completion of the vegetative cycle. Hence, different agronomic strategies based on natural products are being implemented to enhance salinity resilience and improve crop production.

Plant biostimulants (PBs) represent a promising tool in agriculture and provide potential benefits toward the sustainable management of crops by promoting plant growth and nutrition as well as protective effects against environmental stressors [7]. Plant biostimulants can be divided into two major categories. The first one includes bioactive natural substances such as algal extracts, protein hydrolysates, and humic and fulvic acids, whilst the second one comprises beneficial microorganisms, including plant growth-promoting bacteria (PGPB) and arbuscular mycorrhizal fungi [8]. Within the first category, it is worth mentioning plant-derived protein hydrolysates (PHs), which consist of a mixture of polypeptides, oligopeptides and amino acids. These organic compounds have proven themselves capable of enhancing nutrient uptake, promoting root and shoot growth, and alleviating the impact of abiotic stresses, including high salinity [9,10]. The mechanisms by which PHs enhance plant growth may include: (i) the stimulation of carbon and nitrogen (N) metabolism by regulating key enzymes involved in the tricarboxylic acid cycle and N-assimilation pathway, (ii) the production of antioxidant enzymes and metabolites derived from secondary metabolism, and (iii) eliciting auxin- and gibberellin-like activities [9,11].

On the other hand, within the second category of PBs, PGPBs are considered the major approach for enhancing plant growth and overcoming the effects of high salt concentrations [12,13]. PGPB have been shown to induce positive effects on plants, including modification of root system architecture, increased seed germination and seedling development, stimulation of shoot growth, leaf senescence and early blooming, and improvement of fruit formation and grain yield [14,15]. PGPB with these properties have been isolated from diverse ecosystems and belong to several genera, such as *Rhizobium*, *Azospirillum*, *Bacillus*, *Bradyrhizobium*, *Azotobacter* and *Pseudomonas* [7]. These microbial strains exert their activities through a wide variety of mechanisms including hormonal regulation, improvement of nutrients and water use efficiency, cell oxidative balance, and photosynthetic response [16].

Since PBs have emerged as an innovative and promising alternative for mitigating the harmful effects of high salt concentrations in several crops, a deeper understanding of their mode of action is becoming crucial for a better use that maximizes their performances. Yakhin and colleagues [17] pointed out the difficulty in determining the mode of action for PBs and highlighted the importance of demonstrating their positive impact on plant biological processes through careful agronomic experimentation, as well as molecular or biochemical approaches. Hence, metabolomic, proteomic and transcriptomic analyses have become necessary tools to unravel the biochemical and molecular mechanisms triggered in plant–biostimulant–stress interactions [13].

In this context, this work was aimed at (i) identifying the effectiveness of bacterial-based and plant-based biostimulants in ameliorating the resilience of salt-stressed tomato plants and (ii) investigating the changes induced by either PGPB and PHs in the metabolomic profile of both plant tissues and root exudates. We hypothesized that induced tolerance to salinity by PGPB inoculation or PHs application might involve morphological, physiological, biochemical and metabolomic modulations that are specific to the type of biostimulant applied. For this purpose, plants were grown in hydroponics, subjected to saline stress and treated with different biostimulants (i.e., PHs and the PGPB *Azospirillum brasilense*). Physiological and biochemical effects induced by treatments were assessed by analyzing biomass production, root morphology, the activity of antioxidant enzymes and oxidative stress protection. Furthermore, modifications in the metabolomic profile of root exudates (root exudome) and leaves (leaf metabolome) were also investigated by ultra-high performance liquid chromatography/quadrupole time-of-flight mass spectrometry (UHPLC/QTOF-MS).

## 2. Results

### 2.1. Plant Biomass Production and Root System Are Affected by Salinity Stress

High salt concentrations severely affected the growth of both roots and shoots, albeit showing the most evident effect on the aerial part (Figure 1A). In fact, shoot dry weight (SDW) in all salt-stressed treatments was significantly reduced by 33–68% compared to the treatments without salt (Figure 1B). However, the inoculation with *A. brasilense* (Ab) and the application of PHs C (*Malvaceae*-based), D (Trainer^®^) and P (*Poaceae*-based) promoted a significant increase of SDW under high salinity (by 63%, 31%, 31% and 26%, respectively), compared to stressed control plants (Figure 1B). A similar effect was observed in no-salt conditions, where the Ab induced the highest increase of SDW (65%), followed by the plant-derived biostimulants P (50%), D (30%), and H (30%) when compared to control plants (Figure 1B).

Salinity stress also reduced root dry weight (RDW), compared to control plants (Figure 1C). Similarly to SDW, the treatments Ab, C, P and D significantly enhanced the root biomass of salt-stressed plants by 156%, 67%, 66% and 61%, respectively, compared to the corresponding control (Figure 1C). Under no-salt conditions, Ab, P and H treatments promoted higher root biomass accumulation by 118%, 74% and 37%, respectively (Figure 1C). On the other hand, salinization in all the treatments significantly increased the root-to-shoot (R:S) ratio by at least twofold, compared to no-salt conditions (Figure 1D). Additionally, it is noteworthy that tomato plants inoculated with Ab presented a higher R:S ratio under both high-salt and no-salt conditions compared to the corresponding control plants (Figure 1D).

Regarding root morphology, treatment of tomato plants with the bacterial biostimulant Ab significantly enhanced total root length, surface area and volume under both salinity conditions, compared either with their controls or the PHs (Figure 1E–G). However, the application of PHs differentially affected root morphology in both salinity conditions, compared to untreated plants. Under no-salt conditions, H and P were the best performing biostimulants, enhancing length, surface area and volume by 11 and 20%, 20 and 18%, and 15 and 26%, respectively, while under salinity conditions, C, D and P slightly increased length and surface area (Figure 1E,F). On the other hand, the protein hydrolysate H showed a lower effect on these growth traits under salinity, while no significant differences for the root volume (Figure 1E–G).

### 2.2. Total Phenolics and Flavonoids in Leaves

The levels of total flavonoids and total phenolics are provided in Table 1, as mg g^−1^ of fresh leaves. Considering only control plants, it was possible to observe that salt stress induced a significant increase in the content of total phenolics and flavonoids (by 20% and 44%, respectively), compared to non-stressed control plants. However, different responses were observed when plants were treated with biostimulants. Regarding total phenolics content (Table 1), plants treated with the biostimulant C also showed an increased phenolics content under saline conditions, similar to the control. On the other hand, the treatment with biostimulants D and P induced a significant decrease in the content of phenolics in leaves. Meanwhile, no significant differences were detected either in plants treated with Ab or in those treated with the plant-derived biostimulant H.

Regarding the total flavonoid content, albeit the inoculation with Ab induced a decrease by ~20% under saline stress compared to no-salt plants, the inoculation allowed recording the highest levels of flavonoids compared to other biostimulants, regardless of the saline condition (3.3 and 2.6 mg RE g^−1^ FW under no-salt and salt conditions, respectively). As far as the other PHs are concerned, the effects on flavonoid concentrations were variable depending on the treatment, showing a decrease in high salt for D, an increase for H and no significant difference for C and P (Table 1).

### 2.3. Proline Accumulation and Oxidative Stress Protection

The proline accumulation in leaves was remarkably increased in all plants subjected to salt stress (Table 1). Regarding the biostimulant treatments under saline stress, Ab significantly induced the highest accumulation (by 50%) of proline compared to the control plants, whilst the opposite effect was observed in plants treated with the plant-based biostimulants C, D and H. On the other hand, under no-salt conditions, the concentration of proline decreased by 20–60% in plants treated with both types of biostimulants compared to non-biostimulated plants (Table 1).

The stress response of tomato plants to high salt and biostimulant treatments was also evaluated by analyzing the changes in the activity of four antioxidant enzymes (CAT, APX, GPX and SOD) in leaves (Table 1). Two-way ANOVA showed that both salt stress and biostimulant treatments induced significant alterations (*p* < 0.001) in the activity of all the enzymes considered (Appendix A). In general, the antioxidant activity of these enzymes was significantly increased in salt-stressed plants to different extents, according to each biostimulant treatment. The enzymes APX and GPX presented higher activity values ranging from 212–600 µmol H_2_O_2_ min^−1^ mg^−1^ protein and 40–210 µmol guaiacol min^−1^ mg^−1^ protein, respectively; whilst CAT and SOD exhibited lower activities ranging from 3.3–5.4 µmol H_2_O_2_ min^−1^ mg^−1^ protein and 1.5–3.8 units SOD mg^−1^ protein, respectively (Table 1). Under saline conditions, higher CAT activity was observed in the leaves of tomatoes treated with Ab, followed by plants treated with biostimulant D, with a significant increase of 54% and 30%, respectively, compared to the control. However, the highest CAT activity was presented by plants grown in no-salt conditions and treated with Ab. A similar trend was observed for GPX activity, independently from salt stress. Regarding APX activity, it was significantly higher in plants treated with the biostimulant P, regardless of salt conditions, whilst plants treated with D presented the lowest values. The opposite response was observed in SOD activity, where plants treated with the biostimulant P showed lower values, irrespective of the salt conditions; however, plants inoculated with the bacterial biostimulant presented higher SOD activity in leaves.

### 2.4. Metabolomic Profiling of Tomato Leaves and Root Exudates

The characterization of the metabolic profiles of both tomato leaves and root exudates has been performed by untargeted metabolomics to unravel the plant response triggered by the biostimulants considered in our study under either no-salt or salt-stress conditions. The whole datasets, for leaves and root exudates, are provided as Appendix A together with compounds abundance, annotations and composite mass spectra (Appendix A). Firstly, the analysis of the metabolic profiles of samples was performed by using multivariate statistics to elucidate the efficacy of treatments in modulating plant metabolism. Although the biostimulants impacted plant biomass production and root development in the presence/absence of salt, the unsupervised cluster analysis of the metabolomic fingerprints in both leaves and root exudates indicated that the salt stress was the hierarchically prevalent factor in separating the metabolic profiles (Appendix A). Therefore, a supervised model orthogonal projection to latent structures discriminant analysis (OPLS-DA) of the metabolic profiles was carried out, presenting very accurate goodness and fitness parameters (Figure 2A,B).

OPLS-DA simplified the discrimination of metabolic profiles in leaves, showing that H presented the most distinctive profile compared to the control, while Ab and P were similar to non-treated plants under no-salt conditions (Figure 2A). On the other hand, C and D presented similar metabolic fingerprints. However, high salinity amplified the effect of biostimulants compared to no-salt-treated plants. Under high-salt conditions, NaCl (i.e., control plants grown under salt stress) was separated from all the treatments. Moreover, Ab applications separated from NaCl and all PHs, which presented similar profiles under adverse conditions (Figure 2B).

Once confirming the effect of biostimulants on metabolic profiles, the discriminant metabolites that significatively differed from the control, hence explaining the differences observed in plant performance under stress and non-stress conditions following the biostimulant addition, were identified. This analysis (Appendix A) confirmed the evidence suggested by the multivariate statistics (i.e., HCA, OPLS-DA) and revealed that, under no salt, both Ab and P had a slight effect on leaves. At the same time, H, D and C strongly modulated leaf profiles (Figure 2A).

To further understand the implication of these compounds in the plant’s response to the treatments, the entire list was biochemically interpreted using the pathway tool and classified into the biosynthetic metabolic pathways specific to *S. lycopersicum* metabolism. This analysis indicated that the biostimulants modulated several biosynthetic pathways under both salt-stress and no-salt conditions (Figure 2C,D). Under no-salt conditions, C and D presented similar profiles featuring increasing fatty acids and secondary metabolites while decreasing cofactors, amino acids and hormone biosynthesis. Interestingly, the impact on secondary metabolism fluxes depended on the biostimulant applied. For instance, H presented the most distinct fingerprint in secondary metabolism, according to the OPLS-DA (Figure 2A,C). Although Ab and P seemed to have less impact on leaf metabolic profiles, Ab increased the elicited N-containing secondary metabolism, while P increased phenylpropanoids, decreased sterol precursors and had a slight effect on nitrogenated secondary metabolites. The biostimulants C and D promoted the accumulation of glycolipids, while H increased the accumulation of sterols. In this line, H also increased brassinosteroids similarly to C and D, although to a lesser extent. P negatively modulated fatty acids and lipid biosynthesis, while Ab had no significant effects on these biosynthetic pathways.

As observed in the second OPLS-DA score plot, the effect of biostimulants was more pronounced under stress conditions (Figure 2D). In fact, biostimulants triggered several pathways under salinity that differed from those activated in control conditions. This fact suggests a complex interaction between biostimulant treatments and salt toxicity. The Ab inoculation seemed to have much more effect in the presence of NaCl, and its effect on leaf biochemical composition was not only different from the NaCl control but also the other PHs. In addition, when PHs were applied under stress conditions similar profiles were identified between them, in contrast to what was observed under non-stress conditions.

As a general trend, a decrease in amino acids, nucleoside and nucleotide biosynthesis was observed under salt stress, regardless of the treatment. Similarly, fatty acids and lipids biosynthesis, including sterols-related compounds, was particularly modulated (Figure 2D). As a common response to all treatments, phospholipids increased in the presence of NaCl and, in particular, in the presence of NaCl combined with the biostimulants. However, glycolipids increased in the sole presence of PHs under no-salt conditions while digalactosyldiacylglycerol (DGDG) increased in all PH-treatment plants under stress conditions and monogalactosyldiacylglycerol (MGDG) decreased in all NaCl-treated plants. Interestingly, phenylpropanoids were positively modulated by all treatments and, in particular, by all the PHs (Figure 2D; Appendix A). The PH P decreased terpenes under salinity, similar to no-salt conditions. Moreover, α-solanine/α-chaconine biosynthesis, steroidal glycoalkaloids, were strongly elicited in the presence of NaCl in combination with biostimulants rather than in NaCl presence alone. Regarding phytohormones, brassinosteroids increased when Ab, H and P were applied under stress conditions, while cytokinins were regulated in the same manner for all treatments. Furthermore, Ab modulated the superpathway of methylsalicylate metabolism whilst it was not activated by NaCl. Finally, several compounds related to the detoxification of reactive carbonyls in chloroplasts were distinctly modulated by the treatments. For instance, Ab decreased pentanone and hexanal, while C, P and H decreased butanone. In line with plant detoxification, the farnesylcysteine salvage pathway was affected by Ab, P and D by increasing farsenol and farnesyl phosphate, while C decreased oxidized glutathione (Appendix A).

As far as the exudation profile is concerned, Ab-treated plants presented the most distinct pattern of compounds released by the roots under no-salt conditions, as shown after the analysis of the root exudate profiles (Figure 3A). Nevertheless, all the PHs presented a similar metabolic fingerprint in root exudates that were distinct from Ab-inoculated and control plants. The OPLS-DA suggested a milder effect of P compared to the other biostimulants (Figure 3A), and this trend was also confirmed under salt stress (Figure 3B). On the contrary, under saline conditions, P clustered closer to untreated NaCl-stressed plants while Ab separated from the other treatments. Among PHs, H presented the highest effect compared to the NaCl-treated plants (Figure 3B).

Afterward, metabolites distinctly exuded by roots compared to control plants (*p*-value < 0.05; FC > 1.2) were chemically classified using the software ChemRich and the results are depicted in Figure 3C,D. Regarding the specific modulation of exudation by treated plants under no-salt, Ab induced the release of amino acids (glutamate, threonine and citrulline) and phenylpropanoids (anthocyanins, coumarins, flavonoid and caffeic acids), compared to the control (Figure 3C; Appendix A). This trend was the opposite with respect to C, which seemed to repress the exudation. Amino acids (citrulline) and caffeic acids were also exuded by D, H and P. Stilbenes were repressed in all the cases.

In the presence of salinity, the trend was similar between biostimulants (Figure 3D). The inoculation with Ab increased the exudation of amino acids, coumarins, lignans, saponins, and terpenes without affecting the flavonoids’ exudation. On the other hand, PHs negatively affected the exudations of some phenylpropanoids (caffeic acids and anthocyanins) while increasing flavonoids and lignans in the case of D, H and P. More specifically, P elicited the exudation of phenylpropanoids and also carboxylic acids (homoveratric acid and homovanillic acid) (Appendix A).

### 2.5. PAL Gene Expression in Leaves Is Differentially Affected by Biostimulants Treatments

Phenylalanine ammonia lyase (*PAL*; EC 4.3.1.5), is a key enzyme in the phenylpropanoid metabolism pathway [18]. Some members of the *PAL* gene family are well-known stress-responsive genes and their induction is responsible for the synthesis of a wide variety of polyphenolic compounds, including lignin, flavonoid, anthocyanins, cinnamate and phytoalexins [19]. We focused on two members of the *PAL* gene family, namely *PAL2* and *PAL6*, which have already been shown to be regulated in tomato plants under salinity stress [19]. Both genes showed significant up-regulation under high-salt conditions, irrespective of the treatments applied (Appendix A). Regarding the biostimulant treatments under high salt, Ab, D and H induced a lower expression of *PAL2* compared to the saline control, while C and P did not alter the transcription of this gene. A similar trend was observed for *PAL6* expression, yet the PH C induced an overexpression in this case (Appendix A).

## 3. Discussion

Salt stress is an increasing agronomic problem that threatens plant growth, crop productivity and, consequently, food security [20]. Besides the traditional approaches used to cope with high salt concentrations (e.g., water and nutrients supply, conventional breeding), novel practices are being implemented to increase yield and face stress-induced damages [17]. One of these innovative approaches is the use of biostimulants, described as playing a key role in regulating molecular, physiological and biochemical processes in plants that stimulate growth and mitigate the impact of abiotic stresses [9]. Despite gaining increasing relevance in agriculture, the mechanisms underpinning the specific mode of action of plant biostimulants have not been fully elucidated yet. Therefore, to increase our knowledge about the different effects induced by biostimulants in salt-stressed plants, we investigated four plant-based biostimulants (vegetal-protein hydrolysates from different botanical origin) and a bacterial biostimulant (*Azospirillum brasilense*) in tomato plants hydroponically grown under either high-salt (120 mM NaCl) or no-salt (0 mM NaCl) conditions.

A high concentration of NaCl induced a decrease in biomass accumulation (shoots and roots), more evident in the aboveground, in agreement with previous studies [21,22,23,24]. However, as reported by Álvarez and Sánchez-Blanco [25], the root-to-shoot ratio increased under high salinity because the loss of shoot biomass is not mirrored by an equivalent loss of root growth, possibly due to plant adaptive responses aimed at improving water and nutrient uptake. Such changes are supposed to involve the differential modulation of diverse metabolites, osmolytes and phytohormones balance [13,26].

Regarding the use of biostimulants, three out of the four foliarly-applied, plant-derived PHs showed comparable effectiveness in enhancing tomato biomass under high salt (C, D and P) and no salt (D, H and P). Nonetheless, the inoculation with Ab at the root level induced a greater accumulation of tomato biomass in both no-salt and saline conditions. Several works have described the ameliorative effects of vegetal- and microbial-based biostimulants on salt-stressed plants [27,28,29,30]. However, the nature of biostimulants and their mode of application may elicit specific mechanisms of plant growth promotion and stress alleviation [31,32]. For instance, it is well known that root inoculation with *A. brasilense* produces specific secondary metabolites and regulatory molecules that, in turn, induce the development of a more extensive root system, increase the dry biomass, and improve nutrient uptake of plants under multiple environmental conditions [33,34]. On the other hand, despite their mode of action still being unclear, the beneficial effects of the foliar application of protein hydrolysates on plant growth may be due to the action of bioactive compounds (i.e., signaling peptides and amino acids) that induce enhanced biomass, increases in N metabolism, photosynthetic activity and ROS scavenge [35,36].

Although PHs positively affected tomato root development, the bacterial treatment induced a stronger response in both salinity levels. Root morphology and architecture are orchestrated by the interaction among different phytohormones, being the auxin indole-3-acetic acid (IAA) the central modulator [37]. In a previous work, Ceccarelli et al. [28] found that the treatment of tomato plants with PHs shaped the phytohormone profile. Nonetheless, the higher effectiveness of *A. brasilense* in boosting root development in our experiment can be attributed to its greater ability to produce IAA under normal or stressed conditions [38,39]. The auxins produced and secreted by *A. brasilense* are known to stimulate the elongation and differentiation of root cells, the development of lateral and adventitious roots, the number of root hairs, and total length and volume [40,41].

To cope with the oxidative stress induced by high salt concentrations, plants can accumulate osmolytes, such as proline, which play protective roles against oxidative damage by ROS, including stabilization of cell membranes, proteins and photosynthetic apparatus [42]. A higher concentration of proline was recorded in the leaf tissues of salt-stressed plants in our study. However, a differential accumulation of this osmolyte was observed according to the nature of the biostimulant applied. For instance, while the bacterial biostimulant increased the concentration of proline, three out of the four plant-based biostimulants induced a reduction of this osmolyte under high salinity. Plant growth-promoting bacteria have been described to modulate proline expression in plants. For example, the expression of proline biosynthetic gene P5CS1 (∆1-pyrroline-5-carboxylate synthetase 1) was notably up-regulated in *Enterobacter* sp. EJ01-inoculated plants under saline conditions [43] and *Pseudomonas putida*-inoculated plants exposed to drought [44].

In plants exposed to abiotic stress, the treatments with plant-based biostimulants showed contrasting evidence of proline accumulation. For instance, PHs derived from alfalfa were shown to enhance salt tolerance in maize, likely by increasing the accumulation of proline [45], whilst the foliar application of PHs derived from legume seeds and seaweed extracts had no effects on proline content in lettuce plants grown under salinity stress [46]. In this regard, our results underline the distinctive nature of different biostimulants as modulators of specific plant mechanisms for salt stress alleviation. Moreover, some researchers have questioned the conventional hypothesis that “more proline induces better stress tolerance”, suggesting that not just the accumulation of proline but also its simultaneous catabolism is required to maintain plant development under osmotic stress [44,47].

Another important mechanism used by plants to alleviate symptoms of salinity is the activation of the antioxidant enzymatic machinery. The most crucial enzymes known to regulate toxic levels of intracellular H_2_O_2_ are APX, GPX and CAT, while SOD plays important role in the dismutation of O2•^−^ into O_2_ and H_2_O_2_, which is biologically less toxic to the cells [48]. Our findings showed that antioxidant enzymatic activities of CAT, APX, GPX and SOD were enhanced under salinity conditions and responded differently according to the biostimulant applied. For instance, the activities of GPX, SOD and CAT were significantly higher in salt-stressed plants treated with Ab and biostimulant D, while APX activity was more induced in plants treated with biostimulant P, to decrease when biostimulant D was used. Once more, these results suggest that the activation of antioxidative defense in salt-stressed tomato plants depended on the origin and composition of the biostimulant applied since the responses observed were not generalized but rather dependent on the treatment considered. Moreover, our results are in line with other studies, which have also described the role of microbial- and plant-based biostimulants on salt stress tolerance and resistance in plants by activating the antioxidant enzymatic system [49,50,51,52].

On the other hand, the complementary analysis of the metabolic processes derived from the untargeted metabolomics approach revealed that the application of biostimulants in both no-salt and saline conditions activated other antioxidant mechanisms besides enzymatic systems. All biostimulants elicited the plant secondary metabolism, including ROS scavengers in no-salt conditions. Again, the specific metabolic pathways seemed to be distinctively affected in a treatment-dependent manner. However, under stress conditions, plants responded to the biostimulants modulating different pathways from those modulated under non-stress conditions, suggesting a complex interaction when treatments were applied together. Regarding leaves, the most effective biostimulants in mitigating the stress and increasing the biomass did not match those that provoked the highest impact on leaf metabolism. Considering that the elicitation of secondary metabolism could represent a metabolic cost for plants [53], in the case of P and Ab, the elicitation of several pathways of secondary metabolism and detoxification could be enough to alleviate plant stress without a detrimental effect on plant growth and development. However, several responses were common for all the treatments under stress conditions.

Interestingly, all the treatments strongly elicited the superpathway of methylsalicylate metabolism under stress conditions, which was not observed in the NaCl treatment alone. The increase of salicylates and derivatives has been previously observed after applying PHs [36]. Moreover, it is known that *Azospirillum* can synthesize salicylic acid (SA) and other phytohormones. Salicylic acid has an important role in plant response to different stresses, plant tolerance and essential processes such as stomatal closure membrane permeability, photosynthesis, and growth [54]. According to the increase of secondary metabolites we observed, SA is considered a powerful elicitor for all nitrogen-containing secondary metabolites, terpenes and phenylpropanoids [55]. It has been observed that SA is involved in cell redox status [56] and can activate antioxidant enzyme activities and, therefore, reduce oxidative damage [57], which might explain the enhancement of the tomato antioxidant enzyme activities observed in our work. This relates to other important classes of compounds, namely fatty acids and lipids. It has been observed that the increase of SA led to the activation of phospholipase D, key in the biosynthesis of phosphatidic acids [56].

Our study observed lipids reprogramming when biostimulants were applied under stress conditions. In fact, besides phosphatic acids, choline, and derivatives, other important lipidic molecules essential for chloroplast membrane stability (MGDG and DGDG) or sterols that maintain membrane stability and permeability were observed. The alteration of some membrane lipids and MGDG has been observed as a common strategy to increase plant tolerance [58]. Moreover, other phytohormones are connected to lipid modulation. BRs, hormones that increase in the presence of biostimulants and high-salt stress, can modify fatty acid content and affect MGDG, DGDG, and cell lipid composition under salt stress [59].

Besides the leaf status in terms of redox activity and metabolite compositions, our findings revealed that the root exudation was also modulated based on the treatment, salt stress being the main factor in modifying the exudation pattern. In this line, Badri and Vivanco [60] indicated that root exudation is determined by external factors such as biotic and abiotic stressors, among other factors. These external factors can influence both the quantity and quality of root exudates in agreement with our results since we observed changes in exudation abundance and exudates composition compared to the control. Similar findings were found by Zuluaga et al. [13]. These authors revealed that NaCl induced root exudation, including osmotically active compounds, such as carbohydrates and amino acids. In addition, several authors also observed that applying exogenous molecules, such as elicitors, led to stimulating root exudation compared to non-elicited plants [60]. Considering that metabolites exuded by roots are involved in several growth and development processes, in solubilizing nutrients into assimilable forms, and can help plants survive under adverse conditions [61], the distinctive modulation of plant exudates between microorganism inoculation and PHs application may have a role in mitigating salt stress while maintaining plant biomass.

Increased *PAL* activity is commonly associated with the increased production of phenylpropanoid products [62]. In our study, both *PAL* members investigated were increased in tomato leaf tissues in all treatments under saline conditions. Several data sources suggest that overexpression of specific *PAL* gene members can activate the defense mechanisms and promote the functional improvement of plants under abiotic stresses. For instance, salinity stress was reported to induce the upregulation of *PAL6* in the roots of *Medicago sativa* [63] and *PAL1* in leaves of two *Salvia* species, consequently increasing total phenolic accumulation [64]. Thus, our results confirm *PAL* as an indicator of stress conditions in plant species.

## 4. Materials and Methods

### 4.1. Plant-Derived Biostimulants

Four plant-derived biostimulants (protein hydrolysates) were used in this study: C, P, D and H. Two of them were obtained by enzymatic hydrolysis from the botanical families *Malvaceae* (C, *Malvaceae*-based) and *Poaceae* (P, *Poaceae*-based) as previously described [28,65]. The other two (D—Trainer^®^ and H—Vegamin^®^, Hello Nature USA Inc., Anderson, IN, USA) were commercial products resulting from the enzymatic hydrolysis of vegetal-derived proteins. The plant-derived biostimulants were prepared in a concentration of 3 mL L^−1^ of water solution and then evaluated under foliar application mode using a bottle sprayer until the whole aerial part of the plants was covered with a thin layer of liquid, as previously described [66,67]. Plants were treated with plant-derived biostimulants once a week until the end of the trial (Appendix A).

### 4.2. Microbial Biostimulant (Ab)

*Azospirillum brasilense* (DSM-1843) was grown in LB medium (10 g L^−1^ tryptone, 5 g L^−1^ yeast extract, 10 g L^−1^ NaCl) under orbital shaking at 180 rpm, 28 °C for 48 h. After that period, cells were harvested, washed three times, and re-suspended in a sterile saline solution (0.85% *w*/*v* NaCl). Bacterial biostimulant suspension was used to inoculate roots via the hydroponic nutrient solution (root-level application) to a final concentration of 10^6^ cell mL^−1^ in a single dose when plants had grown for 10 days in the hydroponic culture (Appendix A).

### 4.3. Plant Material and Growing Conditions

Tomato (*Solanum lycopersicum* L. cv MicroTom) seeds were germinated for 5 days in the dark at 22 °C on filter paper moistened with 0.5 mM CaSO_4_. After the germination period, the tomato seedlings were transplanted in pots filled with 1.5 L of nutrient solution (NS) containing 0.5 mM MgSO_4_, 2 mM Ca(NO_3_)_2_, 0.1 mM KCl, 0.7 mM K_2_SO_4_, 10 µM H_3_BO_3_, 0.5 µM MnSO_4_, 0.2 µM CuSO_4_, 0.5 µM ZnSO_4_, 0.01 µM (NH_4_)_6_Mo7O_24_, and 80 µM Fe-EDTA. The solution was continuously aerated and renewed twice a week [68]. After eight days of hydroponic culture, plants were split into groups and exposed to two salinity levels: 0 mM and 120 mM NaCl. The salt-stress condition was obtained by supplementing the NS with 7 g L^−1^ of NaCl and maintained until the harvest (Appendix A). Two days after the imposition of salinity, the bacterial and plant-derived biostimulants were applied in the NS or foliarly, as described above. Plants sprayed only with water were used as a control (Appendix A). Three biological replicates (four plants per replicate) were performed for each treatment. Plants were kept under controlled environmental conditions in a climatic chamber 14/10 h light/dark, 24/19 °C, 250 µmol m^−2^ s^−1^ light intensity and 70% relative humidity. After a cultivation period of 21 days, plants were harvested and analyzed as described below. One plant of each biological replicate was used for the collection of root exudates and the assessment of root morphology and plant biomass. The remaining plants of each biological replicate were pooled, their leaves immediately collected and quenched in liquid nitrogen, then stored at −80 °C until use.

### 4.4. Collection of Root Exudates

At the end of the hydroponic cultivation period, one plant of each biological replicate was removed from the nutrient solution and its root system was immersed in plastic pots containing 20 mL of aerated H_2_O MilliQ (18.2 MΩ cm). Pots were covered with aluminum foil to keep the roots in the dark. Root exudates were collected after 4 h of exudation, filtered at 0.45 µm, frozen at −80 °C and freeze-dried. The lyophilized root exudates were then resuspended in 1 mL of water and used for untargeted metabolomics characterization.

### 4.5. Root Morphology and Plant Biomass

For the analysis of root morphology, the whole root system of the same plants used for the collection of exudates was scanned using the WinRHIZOTM system (WinRhizo software, EPSON 1680, WinRHIZO Pro2003b, Regent Instruments Inc., Quebec, QC, Canada). The dry weight of shoots and roots was recorded after drying at 65 °C until constant mass.

### 4.6. Leaf Extraction and Estimation of Total Phenolics and Flavonoids Compounds

An amount of 100 mg of leaf samples, after repeated washing with distilled water, was ground in 1 mL of 80% methanol. The extracts were maintained on ice for 30 min, then centrifuged at 5000× *g* for 30 min at 0 °C. The extracts obtained were filtered through a 0.22 µm syringe filter. One part of the extract was used for the spectrophotometric determination of phenolics and flavonoids, while the other portion was transferred to a vial for metabolomic analysis.

Total phenolics were measured as previously described [69]. Briefly, 200 µL of leaf extract, appropriately diluted, was mixed with 600 µL of water and 200 µL of Folin–Ciocalteau phenol reagent (2 N). After 5 min, 1 mL of aqueous sodium carbonate 7% was added to the mixture and the reaction was allowed to stand in the dark for 60 min at room temperature. The absorbance was measured at 765 nm and gallic acid was used as standard compound. The phenolic content was expressed as mg equivalent of gallic acid per gram of leaf fresh weight.

Total flavonoids were determined according to Chang et al. [70]. Briefly, 200 µL of properly diluted leaf extract was mixed with 600 µL of 95% ethanol, 40 µL of 10% aluminum chloride, 40 µL of 1 M sodium acetate and 1.12 mL of distilled water. The mixture was incubated at room temperature for 40 min, and the absorbance was measured at 415 nm. Rutin was used as standard, and the concentration of flavonoids was expressed as mg equivalent of rutin per gram of leaf fresh weight.

### 4.7. Untargeted Profiling of Root Exudates and Leaf Extracts by UHPLC-QTOF Mass Spectrometry

The untargeted metabolomic profiling of compounds in root exudates (root exudome) and the methanolic extracts of leaves (leaf metabolome) was assessed through a quadrupole-time-of-flight mass spectrometer (Agilent 1290), as previously described [13]. Briefly, analytical conditions were set as follows: 6 μL was injected, chromatographic separation was achieved on a pentafluorophenylpropyl (PFP) column (2.0 × 100 mm, 3 µm—Agilent technologies, Santa Clara, CA, USA) and reverse phase mode binary gradient separation (from 6 to 94% acetonitrile in 33 min) with a flow rate of 200 μL min^−1^ was used. The mass spectrometer was operated in positive SCAN mode (100–1200 *m*/*z*+ range). Profinder B.07 (Agilent Technologies, Santa Clara, CA, USA) was used for features deconvolution, mass and retention time alignment, and filtering (mass accuracy <5 ppm). Compound identification used the completely isotopic pattern (monoisotopic mass, isotope spacing, and isotopes ratio) and the database *Solanum lycopersicum*, version 5.0.1 from PlantCyc 12.6 (Plant Metabolic Network, Hawkins et al. [71], http://www.plantcyc.org; accessed on 20 November 2021) was used for annotation, leading to a level 2 of COSMOS confidence in annotation [72].

### 4.8. Leaf Proline Determination

The proline content in leaf tissues was determined as previously described [73]. Approximately 0.5 g of leaf samples frozen in liquid nitrogen were homogenized in 10 mL of 3% sulfosalicylic acid. The extract was centrifuged at 3000× *g* for 10 min. Two milliliters of freshly-prepared acid-ninhydrin reagent was mixed with 2 mL of the supernatant and incubated in sealed test tubes at 90 °C for 1 h. The mixture was then cooled in an ice bath. The reaction mixture was extracted with 4 mL of toluene and vigorously vortexed for 15 s. The mixture was allowed to stand for 20 min and the chromophore contained in the toluene phase was collected and absorbance was measured at 520 nm in a spectrophotometer. The proline content was estimated from a standard curve and data were expressed as ug proline per g fresh weight (ug g^−1^ FW).

### 4.9. Antioxidant Enzymes Assays

The leaf samples (0.5 g) were frozen in liquid nitrogen and subsequently ground in 5 mL of extraction buffer (100 mM potassium phosphate buffer, pH 7.5, containing 0.5 mM EDTA). The homogenate was centrifuged at 10,000× *g* and 4 °C for 10 min, and the supernatant was collected and immediately used for subsequent analyses of catalase (CAT, EC 1.11.1.6), ascorbate peroxidase (APX, EC 1.11.1.11), guaiacol peroxidase (GPX, EC 1.11.1.7), and superoxide dismutase (SOD, EC 1.15.1.1), as well as for the determination of the total protein content by the Lowry method [74] with bovine serum albumin as a standard.

Catalase activity was determined by following the consumption of H_2_O_2_ at 240 nm. The reaction mixture of 3 mL was prepared by mixing 1.5 mL phosphate buffer (100 mM, pH 7), 0.5 mL H_2_O_2_ (60 mM), 50 µL enzyme extract, and 0.95 mL distilled water. A decrease in the absorbance was measured at 240 nm every 10 s for 2 min. The CAT activity was calculated using the molar extinction coefficient of 39.4 mM^−1^ cm^−1^ and expressed in µmol H_2_O_2_ mg^−1^ protein min^−1^.

Ascorbate peroxidase activity was assessed by following the consumption of ascorbic acid at 290 nm according to Nakano and Asada [75]. Exactly 3 mL of the reaction mixture was prepared by mixing 1.5 mL phosphate buffer (100 mM, pH 7), 0.3 mL ascorbic acid (5 mM), 0.1 mL EDTA (3 mM), 0.1 mL H_2_O_2_ (60 mM), 0.1 mL enzyme extract, and 0.9 mL distilled water. A decrease in the absorbance was assessed spectrophotometrically at 290 nm every 10 s for 2 min. The APX activity was calculated using the molar extinction coefficient of 2.8 mM^−1^ cm^−1^ and expressed in µmol ascorbate mg^−1^ protein min^−1^.

Guaiacol peroxidase activity was estimated by measuring the formation of tetraguaiacol at 470 nm. A volume of 3 mL of reaction mixture consisted of 1.5 mL phosphate buffer (100 mM, pH 7), 0.48 mL guaiacol (100 mM), 0.1 mL H_2_O_2_ (60 mM), 0.1 mL enzyme extract, and 0.82 mL distilled water. An increase in the absorbance was recorded at 470 nm every 10 s for 2 min. The GPX activity was calculated using the molar extinction coefficient of 26.6 mM^−1^ cm^−1^ and expressed in µmol tetraguaicol mg^−1^ protein min^−1^.

Superoxide dismutase activity was determined by measuring the inhibition of blue formazane production in the presence of light according to the method described by Dhindsa et al. [76]. A volume of 0.1 mL riboflavin (60 µM) was added to 3 mL of reaction mixture prepared by mixing 1.5 mL phosphate buffer (100 mM, pH 7.5), 0.1 mL sodium carbonate (1.5 M), 0.2 mL methionine (200 mM), 0.1 mL nitro-blue tetrazolium chloride (NBT) (2.25 mM), 0.1 mL EDTA (3 mM), 0.1 mL enzyme extract, and 0.9 mL distilled water. The tubes were kept for 15 min in a chamber under the illumination of a 15 W lamp. The reaction mixture without the enzyme extract served as the control (development of maximum color), while a non-irradiated complete reaction mixture served as the blank (no development of color). The absorbance was measured at 560 nm. The amount of enzyme required to inhibit 50% of the NBT photoreduction compared to tubes lacking the enzyme extract was defined as one SOD activity unit. SOD activity was expressed on a protein basis, as units mg^−1^ protein.

### 4.10. Gene Expression Analysis

Total RNA was extracted from frozen leaves using the Spectrum Plant Total RNA Kit (Sigma-Aldrich, St. Louis, MO, USA) according to the manufacturer’s instructions. The total RNA (1 μg) was treated with 10U of DNAse RQ1, and cDNA was synthesized using the ImProm-II Reverse Transcription System (Promega, Madison, WI, USA) following the manufacturer’s instructions. Gene-specific primers were designed for the target genes (*PAL2*, NM_001320601.1 and *PAL6*, XM_004249510.4) and the housekeeping gene (elongation factor 1α, NM_001247106.2) (Appendix A). Quantitative real-time reverse transcription PCR (qRT-PCR) was carried out in triplicate as described previously [68]. The relative expression ratio values were calculated by the 2^−ΔΔCt^ method according to Livak and Schmittgen [77].

### 4.11. Statistical Analysis

The results were statistically subjected to two-way ANOVA using R software (version 4.0.3). The mean values were separated according to Tukey’s HSD test with *p* < 0.05 and salinity level effects were compared using the *t*-test. The following R packages were used for data visualization and statistical analyses: ggplot2, agricolae and ggpbur.

The metabolomics-based analyses were performed according to García Pérez et al. [78]. Briefly, raw mass features were elaborated using the software Agilent Mass Profiler Professional B.12.06 for normalization and baselining. Then, the multivariate unsupervised hierarchical cluster analysis (HCA) was performed (Euclidean distance, Ward’s linkage rule) for both leaves and root exudates to describe similarities and dissimilarities among treatments from a fold-change (FC) based heat map. Thereafter, supervised modeling by OPLS-DA was performed in SIMCA 16 software (Umetrics, Sweden). The multivariate models were then cross-validated by Cross Validation-Analysis of Variance (CV-ANOVA, α < 0.01) and fitness and prediction were evaluated by R2Y and Q2Y parameters, respectively.

For leaves, ANOVA and fold-change analysis (*p* value < 0.05, Benjamini-Hochberg multiple testing correction) followed by Tukey post-hoc tests were performed to identify differential compounds between treatments and control. The Omic Viewer Pathway Tool of PlantCyc software (Stanford, CA, USA) was then used for biochemical interpretations [79]. For root exudates, differential metabolites derived from ANOVA and FC analysis (moderate *t*-test, *p*-value < 0.05, Benjamini-Hochberg; FC > 1.5), followed by Tukey post-hoc tests, were classified using the Chemical Similarity Enrichment Analysis (ChemRICH, http://chemrich.fiehnlab.ucdavis.edu/ accessed on 20 November 2021), as previously described [80].

## 5. Conclusions

The results hereby presented demonstrated that *A. brasilense* induced better effects in ameliorating salt tolerance in tomatoes compared to the plant-based PHs biostimulants. Thus, the bacterial-based biostimulant was shown to have a better effect on plant growth, on the accumulation of the osmolyte proline and in the induction of the enzymatic machinery devoted to ROS scavenging. However, it is important to highlight that among the PHs, P (*Poaceae*-based) showed notable effects in counteracting the detrimental-derived effects of salinity in tomatoes. On the other hand, independently of the nature of the biostimulant, plants modulated the metabolome profiles differently in stressing and non-stressing conditions. Nonetheless, the application of biostimulants upregulated antioxidant metabolites in both saline conditions, whilst the superpathway of methylsalicylate metabolism as well as the lipid reprogramming were stimulated under salinity stress. Overall, our observations might indeed suggest that the type of biostimulant (PHs vs. PGPB) and the mode of application (i.e., foliar spray vs. root colonization) can influence the activation of specific pathways in terms of plant tolerance to salinity. To better understand the mode of action of these biostimulants, studies concerning the modulation of the transcriptome, as well as changes in the composition of the rhizosphere microbiome, will be performed.

## Figures and Tables

**Figure 1 ijms-23-14216-f001:**
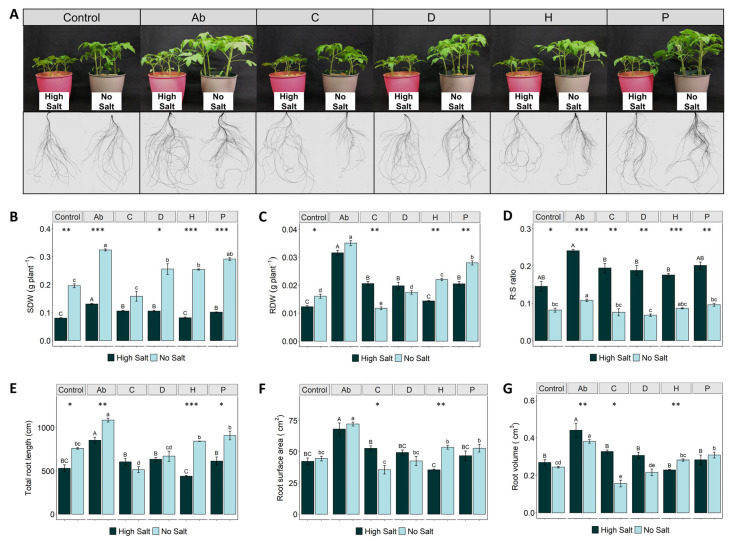
Effects of salt stress (high salt: 120 mM NaCl or no salt: 0 mM NaCl) and biostimulant application (Ab—*Azospirillum brasilense*; C—*Malvaceae*-based; D—Trainer^®^; H—Vegamin^®^; P—*Poaceae*-based; and Control—distilled water) in the biomass accumulation and root morphology of tomato plants growing in hydroponic conditions. Representative pictures of tomato shoots and roots WinRhizo scans (**A**), Shoot dry weight—SDW (**B**), Root dry weight—RDW (**C**), Root to shoot ratio (**D**), Root surface area (**E**), Total root length (**F**), Root volume (**G**). Values are means ± SE. Capital letters compare treatments under high salt and lowercase letters compare treatments under low salt. Equal letters correspond to average values that neither differ according to Tukey’s HSD test (*p* < 0.05). Asterisks indicate significant differences between high and low salt, according to Student’s *t*-test (* *p* < 0.05; ** *p* < 0.01, *** *p* < 0.001).

**Figure 2 ijms-23-14216-f002:**
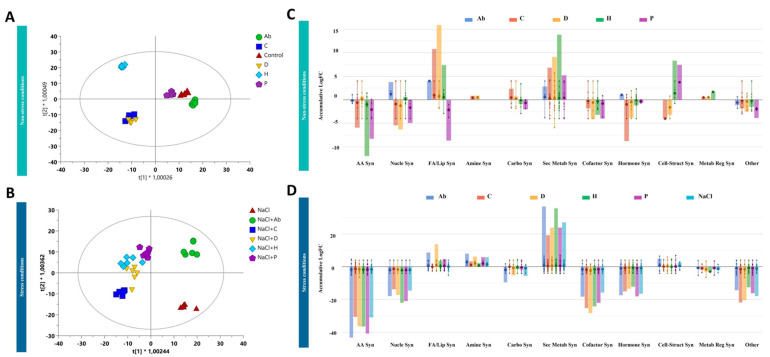
Supervised orthogonal projection to latent structures discriminant analysis (OPLS-DA) score plot built considering the metabolomic profile of tomato leaves under no-salt conditions (R^2^ = 0.99; Q^2^ = 0.8) (**A**) and high salt conditions (R^2^ = 0.96; Q^2^ = 0.8) (**B**). Biosynthetic processes affected by the biostimulants under low salt conditions (**C**) and high salt conditions (**D**). The metabolomic dataset produced through UHPLC-ESI/QTOF-MS was subjected to ANOVA and fold-change analysis (*p* value < 0.05, Benjamini-Hochberg multiple testing correction) followed by Tukey post-hoc tests and differential metabolites were loaded into the PlantCyc Pathway Tool (https://www.plantcyc.org; accessed on 20 November 2021). The large dot represents the average (mean) of all data values for metabolites and the small dots represents the individual logFC for each metabolite. The abbreviated subcategories names on the x axis refer to: AA Syn: amino acids synthesis; Nucleo Syn: nucleosides and nucleotides synthesis; FA/lipids Syn: fatty acids and lipids synthesis; Amine Syn: amines and polyamines synthesis; Carbo Syn: carbohydrates synthesis; Sec Metab Syn: secondary metabolism synthesis; Cofactor Syn: cofactors, prosthetic groups, electron carriers, and vitamins; Hormone Syn: hormone synthesis; Cell-struct Syn: cell structures synthesis; Metab reg Syn: metabolic regulators synthesis; Other: other metabolites.

**Figure 3 ijms-23-14216-f003:**
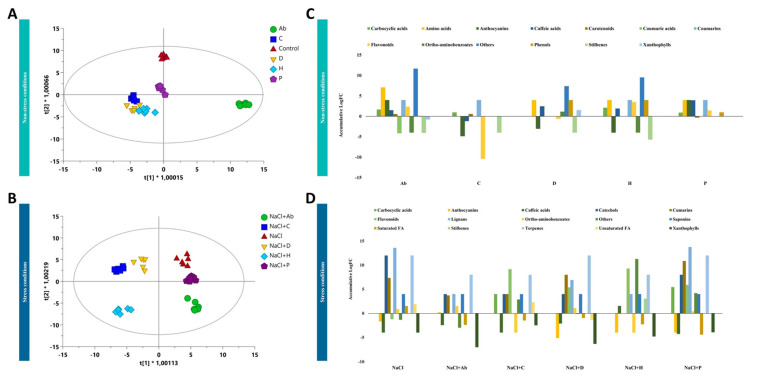
Supervised orthogonal projection to latent structures discriminant analysis (OPLS-DA) score plot built considering the metabolomic profile of tomato root exudates under no-salt conditions (R^2^ = 0.97; Q^2^ = 0.8) (**A**) and high-salt conditions (R^2^ = 0.97; Q^2^ = 0.8) (**B**). Significant compounds exuded by tomato plants treated with biostimulants under no-salt conditions (**C**) and high salt conditions (**D**).

**Table 1 ijms-23-14216-t001:** Effect of biostimulant application and NaCl concentration on phenolics, flavonoids, proline and antioxidant enzymes in leaves of hydroponically grown tomato plants.

Parameters ^a^	Salt Level ^b^	Biostimulant Treatments ^c^					
		Control	Ab	C	D	H	P
Phenolics	High salt	0.42 ± 0.00 Aa	0.45 ± 0.00 a	0.37 ± 0.00 Ab	0.34 ± 0.00 Bc	0.43 ± 0.02 a	0.39 ± 0.02 Bb
	No salt	0.35 ± 0.00 Bc	0.46 ± 0.00 a	0.31 ± 0.00 Bd	0.44 ± 0.01 Aa	0.41 ± 0.00 b	0.45 ± 0.02 Aa
Flavonoids	High salt	1.90 ± 0.04 Abc	2.61 ± 0.06 Ba	1.78 ± 0.02 cd	1.66 ± 0.09 Bd	2.05 ± 0.03 Ab	1.78 ± 0.07 cd
	No salt	1.32 ± 0.06 Bd	3.29 ± 0.16 Aa	1.75 ± 0.02 c	2.67 ± 0.02 Ab	1.95 ± 0.05 Bc	1.77 ± 0.03 c
Proline	High salt	64.74 ± 4.66 Ab	97.73 ± 4.52 Aa	27.28 ± 3.34 Ad	38.74 ± 2.19 Ac	21.32 ± 2.05 Ad	65.60 ± 3.50 Ab
	No salt	1.65 ± 0.148 Ba	1.36 ± 0.06 Bb	0.79 ± 0.07 Bcd	0.67 ± 0.03 Bd	0.97 ± 0.11 Bc	1.00 ± 0.14 Bc
CAT	High salt	3.49 ± 0.11 d	5.37 ± 0.16 Aa	3.81 ± 0.12 Acd	4.53 ± 0.27 Ab	4.09 ± 0.14 Abc	4.11 ± 0.23 Abc
	No salt	3.74 ± 0.41 b	4.27 ± 0.02 Ba	3.39 ± 0.13 Bb	3.31 ± 0.02 Bb	3.53 ± 0.13 Bb	3.32 ± 0.14 Bb
APX	High salt	281.55 ± 20.54 Ade	494.80 ± 50.56 Ab	414.95 ± 6.18 Ac	217.71 ± 10.25 e	345.73 ± 7.72 Bd	600.02 ± 18.09 Aa
	No salt	212.34 ± 15.84 Be	347.37 ± 5.14 Bc	279.21 ± 25.37 Bd	225.71 ± 1.43 e	391.86 ± 2.99 Ab	456.86 ± 22.34 Ba
GPX	High salt	63.61 ± 13.24 d	209.65 ± 5.92 Aa	119.25 ± 15.41 Ac	155.29 ± 2.39 Ab	152.34 ± 15.40 Ab	99.81 ± 2.73 Ac
	No salt	52.49 ± 1.25 b	81.06 ± 5.28 Ba	47.22 ± 12.68 Bb	40.28 ± 1.01 Bb	47.72 ± 2.22 Bb	53.17 ± 3.08 Bb
SOD	High salt	2.28 ± 0.17 Ac	3.78 ± 0.08 Aa	3.01 ± 0.02 Ab	3.39 ± 0.31 Aab	2.57 ± 0.02 Ac	1.63 ± 0.13 d
	No salt	1.57 ± 0.08 Bc	2.52 ± 0.16 Ba	1.76 ± 0.19 Bbc	2.04 ± 0.05 Bb	2.52 ± 0.00 Ba	1.54 ± 0.12 c

Differences between treatments were determined using Tukey’s HSD test and significant differences (*p* < 0.05) are indicated by different lowercase letters when comparing means in rows. Salt level effects were compared using *t*-tests and significant differences (*p* < 0.05) are indicated by different capital letters when comparing means in columns. No significant differences are indicated by omitting notation letters. ^a^ Parameters: phenolics (mg gallic acid equivalent g^−1^ fresh weight), flavonoids (mg rutin equivalent g^−1^ fresh weight), proline (µg g^−1^ fresh weight), CAT (catalase; µmol H_2_O_2_ min^−1^ mg^−1^ protein), APX (ascorbate peroxidase; µmol H_2_O_2_ min^−1^ mg^−1^ protein), GPX (guaiacol peroxidase; µmol guaiacol min^−1^ mg^−1^ protein), SOD (superoxide dismutase; units SOD mg^−1^ protein). ^b^ Salt level: High Salt (nutrient solution supplemented with 120 mM of NaCl); No salt (nutrient solution without NaCl—0 mM). ^c^ Biostimulant treatments: Control, water; Ab, bacterial biostimulant *Azospirillum brasilense*; C, D, H and P, plant-derived biostimulants.

## Data Availability

Not applicable.

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
