# Peer review of "Integrated Metabolomics and Morpho-Biochemical Analyses Reveal a Better Performance of Azospirillum brasilense over Plant-Derived Biostimulants in Counteracting Salt Stress in Tomato"

_ijms, 2022, doi:10.3390/ijms232214216_

Round 1
Reviewer 1 Report
The manuscript is well-written and the figures informative. The metabolomic results are interesting but the transcript expression analysis is poor and do not contribute to the manuscript, indeed it generate more doubts, if possible I will suggest to move these results to supplementary section. Despite this, the manuscript stand on its own merits.
Reviewer 2 Report
Study entitled “Integrated metabolomics and morpho-biochemical analyses reveal a better performance of Azospirillum brasilense over plant-derived biostimulants in counteracting salt stress in tomato” contains novel information which can further strengthen the existing knowledge of the field. Scientists planned their study according to need of time. Moreover, the research area seems well motivated; data presented are sound; data analyses are technically correct and research findings support the claim and objectives properly made in the manuscript. Results are correctly presented and compared with existing knowledge of the field and I think results have some potential broader applicability. Please consider addressing following concerns and incorporate suggestions before any consideration to publish this work.
Minor Concerns
Line 17-20: Too lengthy sentence, rewrite please.
Line 24-27: Too lengthy sentence, rewrite please, how much increase or decrease, present in percentage.
Line 29-32: Too lengthy sentence, rewrite please, how much increase or decrease, present in percentage.
Line 35-36: Mention the keywords in alphabetically orders.
Line 10-104: Rewrite.
Line 505: Remove space between the digit and temperature unit, follow the format throughout the draft.
Line 685: Italicize the p.
Major Concerns
1. Conclude all your results in a quantitative way in the abstract. Language improvement looks necessary as typo mistakes are observed while reviewing the draft.
2. How much quantity/volume of solution was applied per plant and application frequency as well as plant growth stage?
3. Directly state the results with significant findings. Make the results section concise and specific.
4. Try to discuss results with recent literature and provide reasoning of the responses recorded. The introduction and discussion section may be improved by citing the following recent findings;
DOI: 10.35495/ajab.2020.04.219
DOI: 10.35495/ajab.2020.03.164
DOI: 10.35495/ajab.2020.04.257
DOI: 10.35495/ajab.2020.11.556
5. Conclusion may be described with quantification and recommendation.
6. Use journal’s guidelines for the format of references within text and at the end.
Reviewer 3 Report
The research presented by Zuluaga et. al., introduces better performance of Azospirillum brasilense over plant-derived biostimulants in counteracting salt stress in tomato. Current research are based on metabolomics and morpho-biochemical analyses. The manuscipt was well-written.
However, i feel the results could be presented more better, especially for the metabolomics data.
Like enrichment, classication and cluster analysis for related pathway. The metabolomics data in current form is not infomative to readers.
Reviewer 4 Report
The manuscript is well organized and written very well, and before the manuscript is accepted for publication, I have the following comments that should be revised by the author:
1. The key word "abiotic stress" is not selected accurately, and it is recommended to remove or replace it.
2. Some English abbreviations should be given with their full names, such as "TCA-cycle" and "UHPLC-QTOF".
3.In line 119, although the author provides the source of "PHs C, D and P" in the material method section, I suggest the author to explain here to facilitate reader reading; also Figure 1 also needs to be modified.
4. I recommend revising the titles of the results section to be able to profile the results of each section;
5. Line 454-455, the numbers in H2O2 and O2 should be underlabeled.
6. Line 525-526, "both PAL members investigated were overexpressed in tomato leave tissues of all treatments under saline conditions.", the author is not accurate in the expression, it should be that the expression of the PAL gene is increased, and it should not use "overexpressed".
7. Line 608, (SDW) and (RDW) should be deleted;
8. Line 668, Catalase (CAT, EC 1.11.1.6); Line 674, Ascorbate peroxidase (APX, EC 1.11.1.11); Line 682, Guaiacol peroxidase (GPX, EC 1.11.1.7); Line 689, Superoxide dismutase (SOD, EC 1.15.1.1); all shall be marked at line 665.
9. The authors should provide the gene accession numbers for PAL2, PAL6, and the reference genes.
Author Response
Dear Editor and Referee,
We would like to thank you all for your helpful comments and efforts towards improving our manuscript “Integrated metabolomics and morpho-biochemical analyses reveal a better performance of Azospirillum brasilense over plant-derived biostimulants in counteracting salt stress in tomato”. As follows, we report a point-to-point response letter to the concerns raised by the referee, thereby highlighting our efforts to address the criticisms.
RESPONSES TO REVIEWER
1) Reviewer’s comment. The key word "abiotic stress" is not selected accurately, and it is recommended to remove or replace it.
Response: it was replaced by “antioxidant defense system”. We thank for the correction.
2) Reviewer’s comment. Some English abbreviations should be given with their full names, such as "TCA-cycle" and "UHPLC-QTOF".
Response: it was changed. We thank you for the correction.
3) Reviewer’s comment. In line 119, although the author provides the source of "PHs C, D and P" in the material method section, I suggest the author to explain here to facilitate reader reading; also Figure 1 also needs to be modified.
Response: Thanks for the comment. As suggested, we included the vegetal origin of protein hydrolysates in the suggested line, as well as we modified Figure 1 and its legend.
4) Reviewer’s comment. I recommend revising the titles of the results section to be able to profile the results of each section;
Response: We thank the referee for the comment. Wherever possible, we have changed the titles; however, considering that treatments (4 PHs and 1 PGPR) often gave contrasting effects on plants depending on the origin and on the salinity status, we could not change all of them so that they could anticipate the results reported in the paragraph.
5) Reviewer’s comment. Line 454-455, the numbers in H2O2 and O2 should be underlabeled.
Response: it was changed. We thank you for the correction.
6) Reviewer’s comment. Line 525-526, "both PAL members investigated were overexpressed in tomato leave tissues of all treatments under saline conditions.", the author is not accurate in the expression, it should be that the expression of the PAL gene is increased, and it should not use "overexpressed".
Response: We agree with your suggestion and it was changed. We thank you for the correction.
7) Reviewer’s comment. Line 608, (SDW) and (RDW) should be deleted;
Response: it was changed. We thank you for the correction.
8) Reviewer’s comment. Line 668, Catalase (CAT, EC 1.11.1.6); Line 674, Ascorbate peroxidase (APX, EC 1.11.1.11); Line 682, Guaiacol peroxidase (GPX, EC 1.11.1.7); Line 689, Superoxide dismutase (SOD, EC 1.15.1.1); all shall be marked at line 665.
Response: it was changed as suggested. We thank you for the correction.
9) Reviewer’s comment. The authors should provide the gene accession numbers for PAL2, PAL6, and the reference genes.
Response: The gene accession numbers for all genes were provided in the material and method section as suggested.
Reviewer 5 Report
The paper is well written and explained.
Minor English language corrections may be required
Author Response
Reviewer’s comment. The paper is well written and explained. Minor English language corrections may be required
Response: We thank you for your kind comment. A better revision of the English in our manuscript was performed and we expect this point is now covered and corrected.
Round 2
Reviewer 2 Report
Most of the concerns are addressed.
Author Response
Reviewer’s comment: Most of the concerns are addressed.
Response: We thank the referee for the revision of our manuscript.